# Enhanced Expression of the *L1R* Gene of Vaccinia Virus by the tPA Signal Sequence Inserted in a Fowlpox-Based Recombinant Vaccine

**DOI:** 10.3390/vaccines12101115

**Published:** 2024-09-29

**Authors:** Antonia Radaelli, Carlo Zanotto, Chiara Brambilla, Tommaso Adami, Carlo De Giuli Morghen

**Affiliations:** 1Department of Medical Biotechnologies and Translational Medicine, University of Milan, Via Vanvitelli 32, 20129 Milan, Italy; antonia.radaelli@unimi.it (A.R.); chiaraabrambilla@gmail.com (C.B.); tommaso.adami5@gmail.com (T.A.); 2Department of Pharmacy, Faculty of Pharmacy, Catholic University “Our Lady of Good Counsel”, Rr. Dritan Hoxha, 123, 1001 Tirana, Albania; carlo.degiulimorghen@unimi.it

**Keywords:** tissue plasminogen activator signal sequence (tPA), vaccinia virus L1 protein, monkeypox virus, fowlpox virus recombinant vaccines, immunogenicity enhancement

## Abstract

The use of Vaccinia virus (VACV) as a preventive vaccine against variola, the etiological agent of smallpox, led to the eradication of smallpox as a human disease. The L1 protein, a myristylated transmembrane protein present on the surface of mature virions, plays a significant role in infection and morphogenesis, is well-conserved in all orthopoxviruses, and is the target of neutralizing antibodies. DNA recombinant vaccines expressing this protein were successfully used, but they showed lower efficacy in non-human and human primates when used alone, and viral-vectored fowlpox vaccines were already proved to increase immunogenicity when used as a boost. Here, we constructed a novel fowlpox-based recombinant (FP*_tPA-L1R_*), in which the tissue plasminogen activator signal sequence was linked to the 5′ end of the *L1R* gene to drive the L1 protein into the cellular secretion pathway. FP*_tPA-L1R_* expresses a functional heterologous protein that can be immunoprecipitated by hyperimmune rabbit serum. The protein shows cytoplasmic and membrane subcellular localizations and long-lasting expression in CEF, non-human primate Vero and human MRC-5 cells. The tissue plasminogen activator signal sequence can thus contribute significantly to the expression of the L1 protein and may enhance the immunogenicity of a putative DNA/FP *prime–boost* vaccine.

## 1. Introduction

New immunogens against Vaccinia virus (VACV) have been developed with the aim to reduce the possible undesired side effects that may result from the use of traditional vaccines [1]. Recombinant avipox viruses have taken on an important role in the development of novel immunogens. They can infect mammalian cells, but they have an abortive replication cycle, as viral particles are not released, but they can express transgenes for a long time [2,3] and they are able to express foreign antigens for weeks, thus inducing protective immunity in mammals [4,5,6].

As good immune responses are required for vaccine-elicited protection, efforts have been made by stabilizing proteins or by the insertion of sequences with the aim at inducing neutralizing antibodies and protection in animal challenge models [7,8]. To this aim, the tissue plasminogen activator signal sequence (here called tPA, 63 nt) can increase protein expression [9,10] and has recently been used to engineer variants of the spike protein for the construction of the Ad26 vector-based candidate vaccine against SARS-CoV-2 infection [7]. Viral-vectored vaccines are strongly recommended to boost the immune response, which is primed by DNA genetic recombinants [11,12,13,14,15].

Among the identified protective immunogens, the poxvirus transmembrane L1 protein, present on the surface of mature virions (MV) is involved in virus assembly [16] and entry into the cells [17]. It is highly conserved among the OPVXs [18] and shows 98.8% similarity with the M1 MPXV orthologue [19]. It is a major target of potently VV neutralizing antibodies [20], but its expression and immune efficacy may be prevented by its failure to reach the cell membrane [21]. In the presence of tPA, the secreted protein expressed by a viral-vectored recombinant should be transported from the endoplasmic reticulum to the Golgi complex and folded for the correct presentation on the cell surface to the immune system as with DNA-vectored vaccines [22]. The L1 protein, expressed by this novel fowlpox recombinant may thus successfully contribute to enhance the immunogenicity and protection in *prime–boost* immunization regimens against OPVX-related lethal diseases.

Here, we describe the construction and characterization of a recombinant expression plasmid, pVAX*_tPA-L1R_*, and an FP-based recombinant, FP*_tPA-L1R_*, in which the expression of L1 is highly enhanced by the presence of the tPA signal sequence. We demonstrate that FP*_tPA-L1R_* expresses a functional heterologous protein that can be immunoprecipitated by hyperimmune anti-VV rabbit serum. The L1 protein shows both a cytoplasmic and membrane subcellular localization in CEF, non-human primate Vero, and human MRC-5 cells, which is absent when cells are infected with an FP*_L1R_* recombinant, devoid of the tPA signal sequence. RNA transcripts were detected for more than one month, both in human and non-human primate cells. Our further goal will be to combine the administration of the genetic and avipox viral recombinants in *prime–boost* vaccination regimens to increase the response of immunized animals following different *prime–boost* immunization protocols.

## 2. Material and Methods

### 2.1. Cells

Specific-pathogen-free primary chick embryo fibroblasts (CEFs) were grown in Dulbecco’s Modified Eagle’s Medium (DMEM) supplemented with 5% heat-inactivated calf serum (Gibco Life Technologies, Grand Island, NY, USA), 5% Tryptose Phosphate Broth (Difco Laboratories, Detroit, MI, USA), 100 U/mL penicillin, and 100 µg/mL streptomycin (P/S). Human normal lung fibroblast MRC-5 cells and green monkey kidney Vero cells were grown in DMEM supplemented with 10% heat-inactivated calf serum (CS) with P/S.

### 2.2. Expression Plasmid

The expression plasmid pVAX*_tPA-L1R_* was prepared by inserting the *tPA-L1R* sequence into pVAX*zenv* plasmid after removing the *zenv* sequence by the HindIII/NotI restriction enzymes [23]. Briefly, the *tPA-L1R* sequence was amplified by PCR using forward primer V463 (5′ CCC AAG CTT GTC GAC ATG GAT GCA ATG AAG AGA GGG CTC TGC TGT GTG CTG CTG CTG TGT GGA GCA GTC TTC GTT TCG GCT AGC ATG GGT GCC GCA GCA AGC AT 3′) containing the HindIII restriction site and the tPA tissue plasminogen activator signal sequence and reverse primer V464 (5′ CCG GGT ACC GCG GCC GCT CAG TTT TGC ATA TCC GTG GTA G 3′) designed to include the NotI restriction site at the 3′ end. The restriction sequences were needed to clone the *tPA-L1R* sequence into the pVAX HindIII/NotI sites. A PCR reaction was performed with 40 ng of pcDNA3*_L1R_*, 2.5 mM MgCl_2_, 200 μM dNTPs, V463/V464 primers 1 μM, 0.025 U/μL Expand High-Fidelity polymerase (Roche Applied Science, Mannheim, Germany), and nuclease-free H_2_O (Sigma-Aldrich, St. Louis, MO, USA) in a final volume of 20 μL.

The PCR conditions were 94 °C for 2 min, followed by 30 cycles at 94 °C for 30 s, 58 °C for 30 s, 72 °C for 45 s, and extension at 72 °C for 7 min (PTC-200 thermocycler; MJ Research, Waltham, MA, USA). The tPA band was excised from a 1% agarose gel, purified with the Monarch^®^ DNA gel extraction kit (New England Biolabs, NEB, Ipswich, MA, USA). The *tPA-L1R* gene sequence was cut from pcDNA3*_tPA-L1R_* with HindIII/NotI and inserted into the pVAX expression plasmid (Invitrogen Corp., San Diego, CA, USA), cut with the same enzymes and containing the human CMV promoter and approved for use in humans. Transformation was performed using DH5α competent bacteria in the presence of 50 μg/mL kanamicin, as pVAX contains the kanamicin resistance gene. Bacterial selection was performed by PCR amplification using V182 (5′ GGG AAG CTT TTA AAT GGG TGC CGC AGC AAG CAT ACA 3′) and V183 (5′ GGG CTC GAG ATT TTC AGT TTT GCA TAT CCG TGG TAG 3′) primers, starting from 1 μL of each bacterial colony in a final volume of 20 μL, 2.5 mM MgCl_2_, 200 μM dNTPs, and 0.025 U/μL Taq DNA polymerase (Fermentas UAB, Vilnius, Lithuania).The PCR conditions were 94 °C for 4 min, followed by 30 cycles at 94 °C for 30 s, 60 °C for 30 s, 72 °C for 45 s, and extension at 72 °C for 7 min. The DNA was further analyzed both by cutting with HindIII/NotIHF restriction enzymes in *CutSmart* buffer (Fermentas) and running in 1% agarose gel and by sequencing (Bio-Fab Research SrL, Rome, Italy) to exclude the presence of mutations and to verify the correct orientation of *tPA-L1R* into the vector.

### 2.3. Recombination Plasmid

The pFP vector was obtained after deletion of the E7_3G_ gene from pFP_E73G_ recombination plasmid cut with HindIII/NotI^HF^ [24]. The sequence was replaced by the tPA-*L1R* gene excised from the pVAX*_tPA-L1R_* expression plasmid with the same enzymes. Detection of the clone was performed after transformation of DH5α competent bacteria in the presence of 100 μg/mL ampicillin and PCR amplification with 1 μM of each V463/V464 primer, 2.5 mM MgCl_2_, 200 μM dNTPs, 0.025 Taq polymerase U/μL (Fermentas), and nuclease-free H_2_O in a final volume of 20 μL. PCR conditions for the same primers were already described. The selection of the clone was performed after cutting either with HindIII/NotIHF in *CutSmart* buffer or XbaI in *Tango* buffer. Sequencing was also performed to exclude any possible mistakes before amplification on a large scale. Ligation was performed at room temperature by incubating insert and vector at a 5:1 molar ratio for 1 h in the presence of 1.25 U T4 ligase and 5 mM ATP in a final volume of 10 µL with the buffer supplied by the manufacturer (Fermentas). The pFP*_tPA-L1R_* recombination plasmid (9019 bp) contains the tPA signal sequence followed by the *L1R* gene downstream of the VACV H6 (VV_H6_) early/late promoter and was inserted inside the 3-β-hydroxysteroid dehydrogenase 5-delta 4 isomerase gene (DH).

### 2.4. Plasmid Transfection

To overcome the absence of a viral recombinant positive control, transfection was performed using the expression plasmid pWRG*_tPA-L1_*, a kind gift of Jay Hooper (Fort Detrick, MD, USA) that contains the *tPA-L1R* sequence, here called pJH. Transfection was performed on 70–90% subconfluent Vero, MRC-5, and CEF in 5-cm diameter Petri dishes, using lipofectamine (Lipofectamine 3000 transfection kit, ThermoFisher Scientific, Roskilde, Denmark), as per the manufacturer’s instructions. Briefly, solution A, containing 5 µg plasmid DNA and 5 µL P3000 reagent in a final volume of 125 µL DME (no serum and no P/S), was added to solution B, which contains 10 µL lipofectamine in a final volume of 125 µL DME. After a 15 min incubation at room temperature, the medium was removed before adding 1 mL DME and the transfection mixture for 1 h at 37 °C. The complete medium (2 mL/Petri dish) was added for 1–3 days before harvesting the cells.

### 2.5. Fowlpox Virus Recombinant

The FP*_tPA-L1R_* viral recombinant putative vaccine was generated by in vivo site-specific homologous recombination (IVR) between the DH gene of the wild-type FP virus (FPwt, 5 PFU/cell) and the recombination plasmid pFP*_tPA-L1R_*, amplified in CEFs, and purified on discontinuous sucrose density gradient, as already described [23]. The purified virus was resuspended in Ca^2+^-free and Mg^2+^-free phosphate-buffered saline (PBS^−^), briefly sonicated, and then aliquoted and frozen at −80 °C until use. *L1R* gene is thus located downstream of the VV H6 promoter [25] inside the DH gene.

### 2.6. Immunoprecipitation Analysis

Vero, MRC-5, and CEF cells (2 × 10^6^ on 5 cm diameter Petri dishes) were infected with FPwt, FP*_L1R_*, and FP*_tPA-L1R_* (10 PFU/mL) in DME (DMEM met-, cys-, glut-, Sigma, St. Louis, MO, USA) with cys (62.6 mg/L, Sigma), glutamine, and P/S. After 1 h, 20 μCi/mL [^35^S]-methionine (PerkinElmer, Waltham, MA, USA) was added, using the same medium supplemented with 2% dialyzed FCS (Gibco). Cells transfected with pJH were used as a positive control. At 16 h p.i., the cells were washed, harvested, and resuspended in 1 mL lysis buffer (300 mM NaCl, 5 mM EDTA pH 8, 50 mM Tris pH 7.4, 1% Triton X-100, and Na Azide 0.02 mM) with scraping into microcentrifuge tubes, after adding in each Petri dish 0.6 TIU aprotinin (Sigma, St. Louis, MO, USA), 5 µL leupeptin (ThermoFisher Scientific), and 30 µL Calbiochem-Set-1 cocktail protease inhibitor (Calbiochem, La Jolla, CA, USA). The lysate was clarified by centrifugation after a 15 min incubation at 4 °C at 9000× *g* for 15 min at 4 °C. Immunoprecipitation was performed as already described [26] with 2 μL α-L1 polyclonal rabbit serum, kindly provided by BEI Resources (Manassas, VA, USA), previously centrifuged at 800× *g* for 10 min, preadsorbed with the specific cells (Vero or MRC-5 or CEF), and bound to SPA for 4 h to form the complex SPA–Ab, before adding it to the clarified lysate. All incubations were performed at 4 °C. After washing, the pellet was resuspended in Sample Buffer 2x (125 mM TrisHCl, pH 6.8, 4% SDS, 20% glycerol, 10% β-mercaptoethanol, and 0.1% bromophenol blue) before loading. Proteins were resolved using 15% SDS-PAGE and fluorographed.

### 2.7. Expression of Viral RNA Transcripts in Vero Cells by pVAX_tPA-L1R_ and FP_tPA-L1R_

Confluent Vero cells (2 × 10^6^ cells/Petri dish; diameter, 5 cm) were infected either with FP*_tPA-L1R_* or FP*_L1R_* at 5 PFU/cell for 1 h at 37 °C. Vero cells were also transfected with pVAX*_tPA-L1R_* or pVAX*_L1R_*. After 16 h, the cells were rinsed twice with PBS^−^, scraped in 450 μL PBS^-^ from the Petri dishes with a rubber policeman, and centrifuged at 300× *g* for 5 min at room temperature. Cell lysis and RNA extraction were performed according to the QIagen RNeasy mini kits protocol (QIagen, Valencia, CA, USA). RT-PCR was performed using RT-PCR system kits (Access; Promega, Madison, WI, USA) with 25 ng RNA for Vero cells in a final volume of 20 μL in the presence of 1 μM of each primer, 200 μM of each dNTP, 1 U *Thermus filiformis* DNA polymerase, 1 U Avian Myeloblastosis Virus reverse transcriptase, and 1 mM MgSO_4_. *L1R* primers V215 (5′-CAT AGA TGA ATG TTA CGG AG-3′) and V464 were used to obtain a 375 bp band fragment on the *L1R* gene. The *tPA-L1R* specific primers V482 (5′–GTG GAG CAG AGA AGC AAA GC–3′) and V372 (3′–GTG AAC ATA CGC TTG GC–5′) were used to obtain a 655 bp fragment, including a portion of the tPA signal sequence and the *L1R* gene. PCR was performed at 48 °C for 45 min, followed by 2 min at 94 °C for both the couples of primers, and was carried out for 30 cycles at 94 °C for 30 s, 60 °C for 30 s, and 68 °C for 60 s, followed by a final incubation at 68 °C for 7 min.

### 2.8. Over-Time Expression of Viral RNA Transcripts in Vero and MRC-5 Cells

Confluent mammalian replication-restrictive Vero and MRC-5 cells (2 × 10^6^ cells/Petri dish; diameter, 5 cm) were infected in double with FP*_tPA-L1R_* (5 PFU/cell) for 1 h at 37 °C. The cells were harvested for 34 days starting from day 1 after infection and every 3 days for RNA extraction. Primers V482/V372 were planned to include the tPA signal sequence and a portion of the *L1R* gene, using 25 ng RNA for Vero and 75 ng RNA for MRC-5 cells. RNAs from noninfected Vero and MRC-5 cells were also used as positive and negative controls, respectively, as well as cells transfected with pVAX*_L1R_* or with pVAX*_tPA-L1R_*. β-actin was amplified, which gave a band of 518 bp, using 500 ng RNA for Vero cells and 1000 ng for MRC-5 cells in a final volume of 20 μL. The PCR conditions were described above, except for annealing and extension, which were performed at 58 °C for 30 s and 68 °C for 30 s using primers V84 (5′ CTG ACT ACC TCA TGA AGA TCC T 3′ nt 630–651) and V85 (5′ GCT GAT CCA CAT CTG CTG GAA 3′ nt 1147–1127).

### 2.9. Immunofluorescence

Immunofluorescence was carried out as already described [23] using CEFs and Vero and MRC-5 cells to examine the expression and localization of the L1 protein. Briefly, the cells were seeded at a density of 5 × 10^5^/35 mm diameter dish on sterile glass coverslips. Infection was performed with FP*_L1R_* or with FP*_tPA-L1R_* (1–4 PFU/cell, except for CEFs, which were infected with 0.1–2 PFU) at 37 °C for 1 h. After incubation overnight at 37 °C in DMEM supplemented with 2% FCS, the cells were washed twice with PBS^−^. The cells were fixed with 2% paraformaldehyde (Polysciences, Warrington, PA, USA) in PBS^−^ for 10 min at room temperature (for both intracellular and membrane immunofluorescence), followed by 100% cold acetone for 5 min at 4 °C (for intracellular immunofluorescence). The samples were incubated with the 1:50-diluted mouse monoclonal NR-417 anti-L1 serum (BEI Resources, Manassas, VA, USA) or with the 1:50-diluted mouse monoclonal anti-L1 antibody, a kind gift of Jay Hooper (Fort Detrick, MD, USA). The primary antibody was followed by the 1:50-diluted FITC goat anti-mouse antiserum Alexa Fluor Plus (ThermoFisher Scientific). FPwt was used as a negative control. The same procedure was performed to verify the intracellular expression of pVAX*_L1R_* and pVAX*_tPA-L1R_* in transfected CEF cells. The samples were viewed under a fluorescence microscope (Axioskop; Zeiss).

## 3. Results

### 3.1. tPA-L1R Transcript Is Expressed Only by FP_tPA-L1R_

The selected FP*_tPA-L1R_* clone was tested for the expression of its transcript by RT-PCR using the RNA obtained from Vero cells infected with FP*_L1R_* or with FP*_tPA-L1R_*. When *L1R* V215/V464 primers were used, a similar band of 375 bp was shown by the viral (Figure 1A; lanes 1 and 2) and plasmid (Figure 1A; lanes 3 and 4) recombinants either devoid (Figure 1A; lanes 1 and 3) or not (Figure 1A; lanes 2 and 4) of the tPA signal sequence. Conversely, using primers V481/V372 that overlap the TPA signal sequence and the *L1R* gene a 655 bp band was amplified, confirming the presence of the tPA signal sequence only on the two FP*_tPA-L1R_* and pVAX*_tPA-L1R_* recombinants (Figure 1B; lanes 2 and 4). No bands were found when cells were infected with FP*_L1R_* or transfected with pVAX*_L1R_* (Figure 1B; lanes 1 and 3) or in the negative controls (lanes 5).

### 3.2. FP_tPA-L1R_ Expresses the Transgene in Vero and MRC-5 Cells for More than One Month

The expression of the *L1R* transgene after infection by FP*_tPA-L1R_* was also tested over time. The mRNA isolated from the infected Vero and MRC-5 cells showed that the gene carried by FP*_tPA-L1R_* was amplified as a band of 655 bp, which was expressed for up to 34 days p.i. (Figure 2A). In particular, in Vero cells, the expression peaked at 4 days p.i. (lane 2) and remained similar up to 10 days p.i. (Figure 2A, lane 4) to gradually diminish from day 13 (Figure 2A, lane 5) thereafter up to 34 days (Figure 2A, lane 12), after which samples were not harvested any more. In MRC-5 cells, expression peaked at 4 days p.i. (Figure 2B; lane 2), remained similar at day 7 (Figure 2B; lane 3), slightly diminished although maintaining a similar intensity from day 10 up to 19 (Figure 2B; lanes 4, 5, 6, and 7), and was very low from day 22 to 34 p.i. (Figure 2B; lanes 8, 9, 10, and 11). Day 31 could not be harvested. β-actin RNA (518 bp) was similarly amplified in all of the samples (Figure 2A,B), thus confirming the equal levels of total RNA across these different samples. The expression was absent in mock-infected cells (Figure 2A,B), used as negative controls (C).

### 3.3. L1 Protein Is Expressed by FP_tPA-L1R_ in Vero, MRC-5, and CEF Cells

Protein expression was investigated after infection with FP*_tPA-L1R_* recombinant of nonpermissive simian Vero and human MRC-5 cells and permissive CEF using the Western blotting assay. Since no results were obtained and conformational proteins may not be detected by WB, immunoprecipitation was performed after infection with FP*_tPA-L1R_* to bind antibodies to native proteins before loading them onto the gel. Thus, after protein labelling with ^35^S L-Met, the L1 protein, tied to the specific rabbit polyclonal antibodies, was revealed as a dimer of 25–27 kDa (Figure 3; lanes 4) [22]. The same bands were also present when CEF were transfected with pJH, which was used as a positive control (Figure 3; lane 5). When the cells were infected with the previously constructed FP*_L1R_* recombinant, used as a comparative control (Figure 3; lanes 3), very faint bands were found only in Vero cells. As expected, no specific bands were present in the mock-infected cells (Figure 3; lanes 1) or in the cells infected with FP wild type (Figure 3; lanes 2).

### 3.4. L1 Protein Is Expressed by All the Cell Types in the Cytoplasm and on the Membrane by FP_tPA-L1R_

To determine the subcellular localization of the L1 protein expressed by FP*_tPA-L1R_*, the CEFs and Vero and MRC-5 cells were infected with FP*_tPA-L1R_* and analyzed by immunofluorescence (Figure 4A). The assays were performed both after fixation and permeabilization of the cells for intracellular immunofluorescence (row c) or after fixation for membrane immunofluorescence only (row d). These data show that the L1 protein was expressed both at the cytoplasmic level (intracellular IF, Figure 4A(1c,2c,3c)) and all over the membrane (membrane IF, Figure 4A(1d,2d,3d)). The intensity of the fluorescence was similar in the different kinds of cells but was not present in cells infected with the FP*_L1R_* (Figure 4A(1b,2b,3b)), where the recombinant was lacking the tPA signal sequence. The FPwt-infected cells were always negative, as expected (Figure 4A(1a,2a,3a)). The expression of pVAX*_L1R_* and pVAX*_tPA-L1R_* recombinant plasmids was also compared on CEFs. A very low intracellular expression was observed by pVAX*_L1R_* (Figure 4B(a)), whereas high specific immunofluorescence was shown by the pVAX*_tPA-L1R_* recombinant plasmid provided with the tPA signal sequence both after intracellular and membrane fixation (Figure 4B(b,c)).

## 4. Discussion and Conclusions

Safe vaccines against OPXV infection, which affects both animals and humans, are still an important issue [27,28,29]. This has been encouraged by the increased cases of human MPXV zoonotic and inter-human infections [30], by the reduction in ‘herd immunity’ following the discontinuation of the smallpox vaccination campaign, and by problems that may arise from a deliberate release of pathogenic orthopoxviruses (OPXVs) for terrorist purposes [31,32].

Engineered viral vectors and their combination with genetic vaccines can significantly increase immune responses and influence vaccine efficacy [33]. To this purpose, the tPA signal sequence was coupled to the 5′ end of the *L1R* gene of both the viral and DNA recombinants with the aim at providing new immunogens to enhance the L1 protein expression of a putative OPVX vaccine.

Here, we have demonstrated that (i) the FP*_tPA-L1R_* recombinant can express the *L1R* transcript for more than one month in non-human primate Vero and human MRC-5 cells, in which the FP vector does not replicate; (ii) the L1 foreign protein can be immunoprecipitated by hyperimmune serum and shows both cytoplasmic and membrane subcellular localizations; and (iii) the tPA signal sequence inserted ahead of the *L1R* gene significantly contributes to the expression of the protein and can thus be used to enhance immunogenicity in viral-vectored recombinants.

FP*_tPA-L1R_* transcript expression of the heterologous protein was demonstrated by using primers overlapping both the short tPA signal sequence and L1R. This allowed us to confirm the presence of tPA and exclude the amplification of *L1R* only, the sequence of which was used as a probe to identify the clones produced by in vitro homologous recombination (IVR). Over time, transgene expression in replication-restrictive Vero and MRC-5 cells was similar up to day 19 p.i. and was maintained at high levels up to day 13 p.i. Transcript expression enhancement was still present at day 34 p.i. in both cell lines, although higher in Vero cells, probably because of the different animal origin of the two cell lines and the higher replicative potential of the virus in Vero cells. This long-lasting expression suggests that a good immune response may be obtained by this viral recombinant in mammalian cells.

The heterologous VV-specific L1 protein was detected by the immunoprecipitation assay in cells infected with the FP*_tPA-L1R_* recombinant, but not by the Western blot assay. The expression was similar in replication-permissive CEFs and in replication-restrictive Vero and MRC-5 cells and was obtained only after adopting strategies, including the use of several protease inhibitors and low-temperature incubations. Using WB, denatured proteins are loaded on the gel so that the antibody, which is used thereafter, may not recognize conformational proteins. Therefore, it is not uncommon that proteins undetectable by WB can be revealed by IP. Conversely, using IP, proteins are bound to antibodies before loading them onto the gel so that native proteins can be recognized in their native conformation. This cleaner and more sensitive assay, although requiring a careful set up, favors specific binding and was able to reveal a dimer of 25 and 27 kDa. This dimer was also detected after transfecting CEF cells with plasmid pJH, expressing the L1 protein, and used as a positive control. Our findings are, therefore, in line with the conformation-specific binding of antibodies that recognize discontinuous epitopes present on the loops joint by disulfide bonds [34]. In the L1 envelope myristylated protein, these bonds, formed in the cytoplasm by the virus-encoded disulfide bond formation pathway [22,35], may be critical for the antibody recognition of the protein. Conversely, the heterologous L1 protein expression by FP*_L1R_*, in which the tPA signal sequence is not present, was very low or completely absent in the different cell lines, which underlines the importance of the tPA signal sequence. The native L1 is not driven in the cellular secretion pathway, provided with the glycosylation machinery and allowing the correct folding of the protein [36], necessary for antibody recognition and virus neutralization. L1 detection by immunoprecipitation may thus be ascribed to the recognition of the nondenatured form of the protein by the used polyclonal rabbit hyperimmune serum.

By immunofluorescence, the L1 protein was also well expressed in all of the cell lines by either the FP*_tPA-L1R_* or pVAX*_tPA-L1R_*. Fluorescence was localized both in the cytoplasm and on the cell surface, but it was almost undetectable when cells were infected with either the FP*_L1R_* or transfected by the pVAX*_L1R_* recombinant constructs, in which the tPA signal sequence was absent.

Overall, the tissue plasminogen activator signal sequence linked to viral-vectored genes can facilitate the transport of the expressed protein to the cell membrane and enhance the immune response against mammalian OPVXs. This fowlpox-based immunogen can thus be used as a boost after priming with a DNA recombinant expressing the same gene to improve vaccine efficacy in humans in a *prime/boost* immunization regimen. Preliminary in vivo results indicate an increased humoral immune response due to L1, the expression of which was enhanced by the presence of the tPA signal sequence ahead of the *L1R* gene (personal communication). The use of tPA to enhance protein expression and, consequently, the immune response can thus be pursued also for the other proteins.

## Figures and Tables

**Figure 1 vaccines-12-01115-f001:**
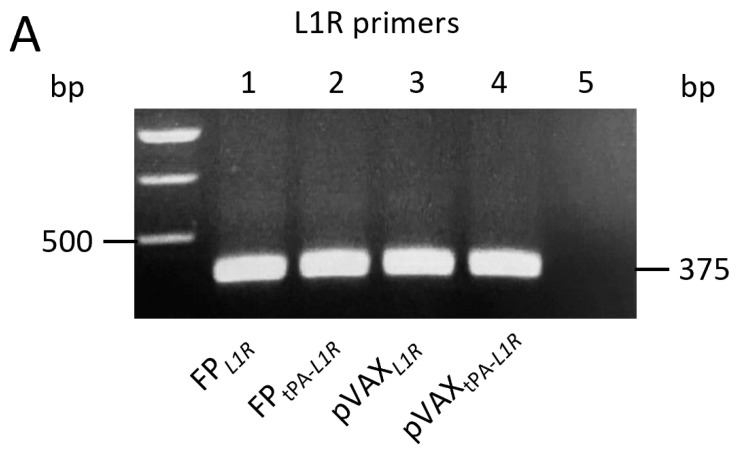
Expression of the *L1R* transcripts by FP*_tPA-L1R_* and pVAX*_tPA-L1R_* recombinants in Vero cells. Cells were infected with the FP or transfected with pVAX recombinants, bearing the *L1R* or the *tPA-L1R* gene sequences. The mRNA expression of the *L1R* region was determined by using two different set of primers. (**A**) The first set of primers, spanning the *L1R* region, shows the expression of the mRNA transcripts by all of the immunogens, either viral (panel A; FP infected cells, lanes 1 and 2) or genetic (panel A; pVAX transfected cells, lanes 3 and 4) recombinants in the absence (lanes 1 and 3) or in the presence (lanes 2 and 4) of the tPA signal sequence. (**B**) The second set of primers, overlapping the tPA signal sequence and a part of the *L1R* sequence, amplified only transcripts obtained after infection with FP*_tPA-L1R_* or transfection with pVAX*_tPA-L1R_* recombinants (panel B; lanes 2 and 4), as expected. No bands were found when using the FP*_L1R_* or the pVAX*_L1R_* constructs (panel B; lanes 1 and 3) or in mock-infected cells (lane 5).

**Figure 2 vaccines-12-01115-f002:**
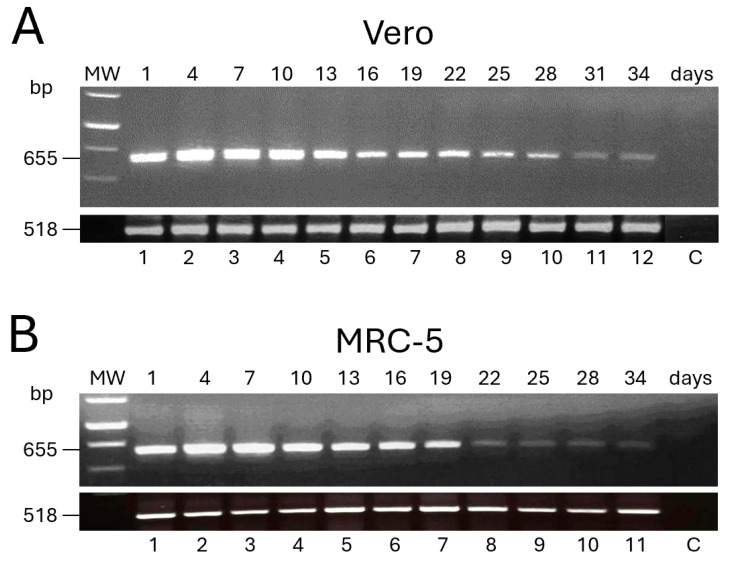
Expression of the *L1R* transcripts over time by FP*_tPA-L1R_* recombinant in replication-restrictive mammalian cells. The expression of the transgene was evaluated by RT-PCR every 3 days, for 34 days, after infection of Vero (**A**) and MRC-5 (**B**) cells with FP*_tPA-L1R_*. Both Vero and MRC-5 cells show 655 bp *L1R* transcripts up to day 34 p.i. A similar trend of expression was found up to day 19 (**A** vs. **B**), whereas a higher expression was seen in Vero cells than in MRC-5 from day 22 to 28. Amplification of human β-actin RNA (518 bp) is shown. MW; 1 kb ladder.

**Figure 3 vaccines-12-01115-f003:**
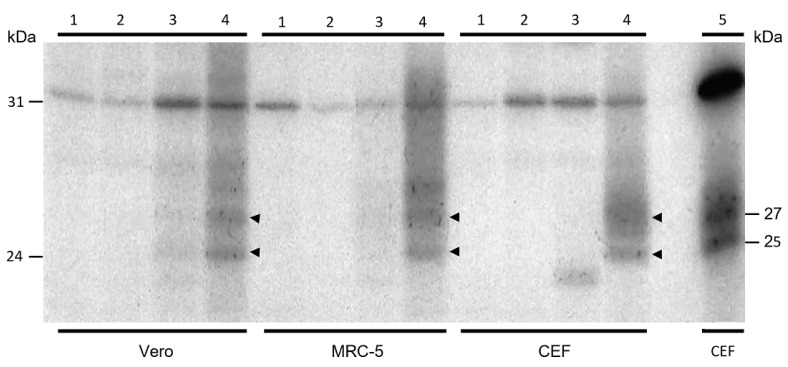
Immunoprecipitation of L1 protein from lysates of Vero, MRC-5, and CEFs. Labelled proteins from lysates of Vero, MRC-5, and CEF infected cells were immunoprecipitated with preadsorbed α-L1 polyclonal rabbit serum and resolved on a 15% SDS-PAGE gel. Cells were mock-infected (lanes 1) or infected with the FP wild type (lanes 2), with the FP_L1R_ recombinant (lanes 3), or with the FP*_tPA-L1R_* recombinant (lanes 4). The different cell lines are indicated at the bottom of the figure. Molecular weight markers are indicated. The rabbit polyclonal antibodies α-L1 revealed the L1 protein as 25–27 kDa bands (lanes 4) that were present also when CEF were transfected with pJH, used as a positive control (lane 5). After infection with the FP*_L1R_* recombinant, very faint bands were found only in Vero cells (lanes 3). No specific bands were present in the mock-infected cells (lanes 1) or in the cells infected with FP wild type (lanes 2).

**Figure 4 vaccines-12-01115-f004:**
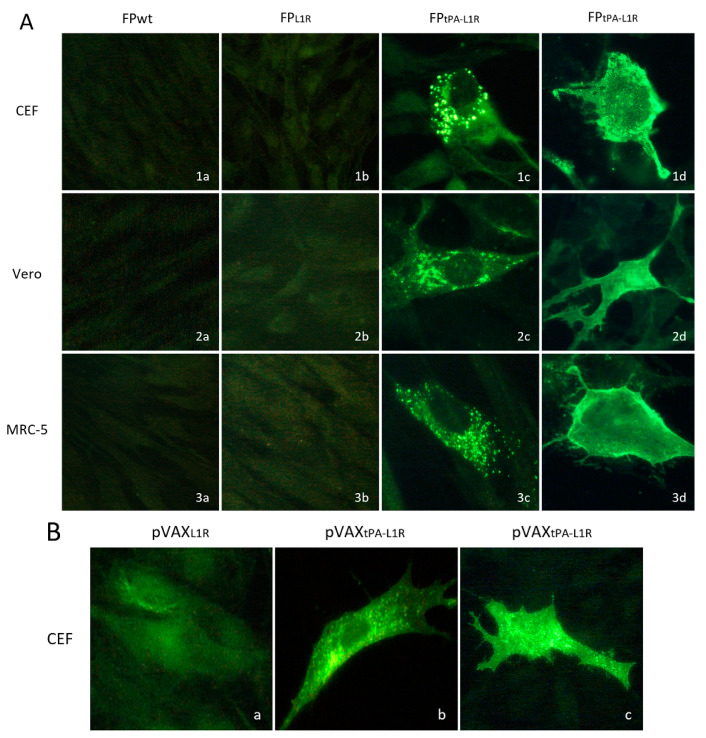
Heterologous protein expression and localization by immunofluorescence in the CEFs and the Vero and MRC-5 cells. Immunofluorescence of the infected cells was performed to determine the subcellular localization of the L1 protein expressed by FP*_tPAL1R_* and by pVAX*_tPA-L1R_*. (**A**) In all of the cell types, either replication-permissive (CEFs; row 1) or nonpermissive (Vero, MRC-5; rows 2 and 3, respectively), the intensity of the fluorescence signal shown by FP*_tPA-L1R_*-infected cells was high (**1c**, **2c**, **3c**, **1d**, **2d**, and **3d**). Conversely, the immunofluorescence shown by FP*_L1R_*-infected cells was very low (**1b**, **2b**, and **3b**). The mouse monoclonal anti-L1 serum detected the L1 protein both in the cytoplasmic compartment and all over the cells. Intracellular and membrane immunofluorescence were shown by all of the cell lines (**1c**, **2c**, **3c**, **1d**, **2d**, and **3d**). No immunofluorescence was detected in the FP-wild-type-infected cells used as negative controls, as expected (**1a**, **2a**, and **3a**). (**B**) CEF transfected with pVAX*_tPA-L1R_* also show higher immunofluorescence than cells transfected with pVAX*_L1R_*, both after intracellular (**b** vs. **a**) or membrane immunofluorescence (**c** vs. **a**).

## Data Availability

The authors make the data supporting their findings available upon request to the corresponding or to the first author.

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
