# Peer review of "Enhanced Expression of the L1R Gene of Vaccinia Virus by the tPA Signal Sequence Inserted in a Fowlpox-Based Recombinant Vaccine"

_vaccines, 2024, doi:10.3390/vaccines12101115_

Round 1

Reviewer 1 Report

Comments and Suggestions for Authors

Dear authors

I hope this finds you all well. Regarding the review of manuscript number vaccines- 3175396, entitled "Enhanced Expression of the L1R Gene of Vaccinia Virus by the tPA Signal Sequence Inserted in a Fowlpox-Based Recombinant Vaccine ". It is a really interesting study that provides important points regarding the expression of the L1R gene in a Fowlpox-based recombinant vaccine; however, some minor comments should be answered.

1-    In discussion, line 393: please indicate how the FPtPA-L1R recombinant can express the L1R transcript for more than one month in non-human primate Vero and human MRC-5 cells, in which the FP vector does not replicate??

2-    Please discuss why the heterologous VV-specific L1R protein was detected by the immunoprecipitation assay in cells infected by the FPtPA-L1R recombinant, but not by the Western blot assay???

Author Response

Dear authors

I hope this finds you all well. Regarding the review of manuscript number vaccines- 3175396, entitled "Enhanced Expression of the L1R Gene of Vaccinia Virus by the tPA Signal Sequence Inserted in a Fowlpox-Based Recombinant Vaccine ". It is a really interesting study that provides important points regarding the expression of the L1R gene in a Fowlpox-based recombinant vaccine; however, some minor comments should be answered.

1-    In discussion, line 393: please indicate how the FPtPA-L1R recombinant can express the L1R transcript for more than one month in non-human primate Vero and human MRC-5 cells, in which the FP vector does not replicate??

FP recombinants enter mammalian cells, but have an abortive replication cycle, as viral particles are not produced, but that they can express transgenes also for a long time. A comparative study with another frequently used avipox virus (Canarypox, CP, ALVAC) was published to compare the two avipox viruses (Zanotto et al. 2010, Antiviral Res 88). We modified a sentence in the Introduction section to make it clearer.

2-    Please discuss why the heterologous VV-specific L1R protein was detected by the immunoprecipitation assay in cells infected by the FPtPA-L1R recombinant, but not by the Western blot assay???

The immunoprecipitation assay (IP) is a cleaner, more advanced and valuable assay than the Western Blot (WB) assay. By WB, proteins loaded on the gel are denatured, whereas by IP the binding of SPA-Ab to the protein occurs before loading the protein unto the gel, as described in the Material and Methods section. Thus, by IP also native proteins are loaded, recognized also for their conformation. IP thus allows the detection of both denatured and non-denatured proteins. However, IP assay requires the use of radioactivity and longer times, it needs to be carefully set up, and therefore it is not used routinely.

Consequently, proteins that may not be recognized by WB, can be recognized by IP. Thus, as the heterologous L1 protein was detected only by IP in the three cell lines infected by the FPtPA-L1R recombinant, this suggests that:

  1. the protein is expressed (which is the aim of our study) not only by CEF where the virus replicates, but also by non-human primate cells (Vero) and human MRC-5 cells, where an abortive replication cycle takes place
  2. if the protein is not detected by the WB assay, this may be due to denatured proteins
  3. in our case, both mouse monoclonal and rabbit polyclonal antibodies were used in WB unsuccessfully. Obviously, we did not show the negative results of our WBs, but shifted to the IP assay, which could give us the chance to detect native proteins, recognized by their conformation
  4. the L1 detection by IP may thus be ascribed to the recognition by the polyclonal rabbit hyper-immune serum of the native conformational form of the protein, which was run in the IP assay, whereas the denatured form of the protein, which was run in the WB assay, could not be detected.

We explained it better in the Discussion and Conclusions section.

Reviewer 2 Report

Comments and Suggestions for Authors

The authors describe a system to express the viral L1R protein of vaccinia virus as a fusion with tissue plasminogen activator signal sequence (tPA) and demonstrate the effect of the tPA sequence to enhance protein expression from a FowlPox based system. Although the authors have provided evidence to support the claim that tPA sequence aids in protein expression of L1R, some critical aspects in support of the claim is lacking.

Specific comments to resolve the shortcomings and better support the authors claims:

1.     Although the authors show by immunoprecipitation the expression of FPtPA-L1R protein to be dependent on the presence of tPA sequence, identifying the immunoprecipitated protein to be tPA-L1R by western blotting would be more definitive. Probing the immunoprecipitated protein by western blotting using anti-tPA antibody and or L1R antibodies would be ideal. The authors claim that L1R antibody is not effective to detect the protein by western blotting, but do not provide any evidence to support this. An alternative would be to use commercially available anti-tPA antibody.

2.     In the data presented in Figure 3, how was the protein normalized for input in the immunoprecipitation assay? What is the non-specific band that runs close to 31 kDa? There appears to be multiple diffused bands in the range of 21-27 kDa. What are the two bands indicated by the arrowheads, the authors do not mention or refer to them in a any specific way in the results. Are they L1R isoforms or some other non-specific proteins that are pulled down in the IP? In this regard, using the pTA antibody in a western blotting assay will be helpful in identifying isoforms of L1R, if any.

3.     Although the tPA sequence appears to enhance L1R protein expression from FPpTA-L1R compared to FP-L1R, the authors have not explored to what extent does tPA sequence enhance (fold enhancement) expression of L1R protein? A quantitative measurement of expression levels of L1R protein from FP-L1R vs FPtPA-L1R will be highly informative and may be helpful in the next steps in devising a successful vaccine.

4.     The RNA expression wanes substantially after day 19 pi. What is the status of protein expression? Does the protein expression mimic RNA expression, or does it wane before? This might be valuable information as the proposal is to develop a vaccine and it will be directly related to the level of antigen expressed and presented for an immune response. A time course of protein expression by western blotting using anti-tPA antibodies will be insightful.

5.     Presenting an image of a single cell in support of the hypothesis that tPA sequence is essential for expression of L1R and its localization to the membrane is hardly sufficient. Including a field of view with multiple cells and the enlarged single cell image will be more appropriate.

6.     Figure 1 A & B, label lane named “C” as 5. As labeled, it is confusing. Correct the labeling on line 295.

7.     Legend to Figure 3. Line 35: shouldn’t this be (lane 4) and not lane 5?

8.     Line 362: (Fig. 4A; 1a, 1b, 1c). Shouldn’t this be 1a, 2a and 3a?

9.     Figure 4: “A” missing from Figure number.

10.  Lines 311 and 314: expression “peaked” at 4 days instead of “raised”.

Author Response

The authors describe a system to express the viral L1R protein of vaccinia virus as a fusion with tissue plasminogen activator signal sequence (tPA) and demonstrate the effect of the tPA sequence to enhance protein expression from a FowlPox based system. Although the authors have provided evidence to support the claim that tPA sequence aids in protein expression of L1R, some critical aspects in support of the claim is lacking.

Specific comments to resolve the shortcomings and better support the authors claims:

  1. Although the authors show by immunoprecipitation the expression of FPtPA-L1R protein to be dependent on the presence of tPA sequence, identifying the immunoprecipitated protein to be tPA-L1R by western blotting would be more definitive. Probing the immunoprecipitated protein by western blotting using anti-tPA antibody and or L1R antibodies would be ideal. The authors claim that L1R antibody is not effective to detect the protein by western blotting, but do not provide any evidence to support this. An alternative would be to use commercially available anti-tPA antibody.

By WB the L1 protein could not be detected even by infecting cells with the new FPtPA-L1R recombinant. This is the reason why we decided to immune-precipitate the L1 protein and verify whether it was expressed, if it was expressed better than L1 protein from the unmodified FPL1R recombinant (in which tPA was absent), if it was differently expressed by the different cell lines.

The immunoprecipitation assay (IP) is a cleaner, more advanced and valuable assay than the Western Blot (WB) assay. By WB, proteins loaded on the gel are denatured, whereas by IP the binding of SPA-Ab to the protein occurs before loading the protein unto the gel, as described in the Material and Methods section. Thus, by IP also native proteins are loaded, recognized also for their conformation. IP thus allows the detection of both denatured and non-denatured proteins. However, IP assay requires the use of radioactivity and longer times, it needs to be carefully set up, and therefore it is not used routinely.

Consequently, proteins that may not be recognized by WB, can be recognized by IP. Thus, as the heterologous L1 protein was detected only by IP in the three cell lines infected by the FPtPA-L1R recombinant, this suggests that:

  1. the protein is expressed (which is the aim of our study) not only by CEF where the virus replicates, but also by non-human primate cells (Vero) and human MRC-5 cells, where an abortive replication cycle takes place
  2. if the protein is not detected by the WB assay, this may be due to denatured proteins
  3. in our case, both mouse monoclonal and rabbit polyclonal antibodies were used in WB unsuccessfully. Obviously, we did not show the negative results of our WBs, but shifted to the IP assay, which could give us the chance to detect native proteins, recognized by their conformation
  4. the L1 detection by IP may thus be ascribed to the recognition by the polyclonal rabbit hyper-immune serum of the native conformational form of the protein, which was run in the IP assay, whereas the denatured form of the protein, which was run in the WB assay, could not be detected
  5. the use of commercial antibodies against tPA would not solve the problem. We inserted in our recombinant only the tPA signal sequence (a very short sequence) which is cleaved away in the mature protein. Therefore, anti-tPA antibody could not be used to detect the L1 protein. Moreover, by using an anti-tPA antibody, we would not detect the L1 protein expressed by the unmodified FPL1R recombinant, used for comparisons in lanes 3.
  6. the L1 myristylated envelope protein contains disulfide bonds, formed in the cytoplasm by the virus-encoded disulfide bond formation pathway (Kou et al, Immunol Lett 190, 2017; Ondondo et al. J HIV & AIDS 2.4, 2016), that may be critical for antibody recognition of the protein.

We tried to better explain it that in the Discussion and Conclusions section.

  1. In the data presented in Figure 3, how was the protein normalized for input in the immunoprecipitation assay? What is the non-specific band that runs close to 31 kDa? There appears to be multiple diffused bands in the range of 21-27 kDa. What are the two bands indicated by the arrowheads, the authors do not mention or refer to them in a any specific way in the results. Are they L1R isoforms or some other non-specific proteins that are pulled down in the IP? In this regard, using the pTA antibody in a western blotting assay will be helpful in identifying isoforms of L1R, if any.

As described in the Material and Methods section, infection with the recombinants was performed on the same number of infected cells (i.e. 2 x 106 on 5 cm diameter Petri dishes). Thus, the amount of proteins used for IP should be almost the same. We did not care about the band close to 31 kDa, because spurious bands are often present (especially by WB) and because this band is also visible in the negative controls (uninfected cells in lanes 1 or cells infected with the FP wild type virus in lanes 2). Conversely, the two bands of 25-27 kDa indicated by the arrowheads correspond to the L1 dimers, also found by other researchers, as discussed in the Discussion and Conclusions section. They also correspond to the dimer found in CEF transfected with pJH, which was used as a positive control and absent in the negative controls. L1 detection by immunoprecipitation may thus be ascribed to the recognition of the non-denatured form of the protein by the polyclonal rabbit hyper-immune serum. We better clarified these points in the Results and Discussion and Conclusions section.

  1. Although the tPA sequence appears to enhance L1R protein expression from FPpTA-L1R compared to FP-L1R, the authors have not explored to what extent does tPA sequence enhance (fold enhancement) expression of L1R protein? A quantitative measurement of expression levels of L1R protein from FP-L1R vs FPtPA-L1R will be highly informative and may be helpful in the next steps in devising a successful vaccine.

The enhancement of protein expression is very clear both by IP and by the IF assay, but a quantitative comparison of the protein bands cannot be performed, as bands are almost undetectable in lanes 3 by any of the cell lines. The low expression level of L1 by the FPL1R recombinant was studied in vivo previously: it did result in a very poor antibody response (Bissa et al, Antiviral Res 134, 2016) and suggested the construction of a this new FPtPA-L1R recombinant.

  1. The RNA expression wanes substantially after day 19 pi. What is the status of protein expression? Does the protein expression mimic RNA expression, or does it wane before? This might be valuable information as the proposal is to develop a vaccine and it will be directly related to the level of antigen expressed and presented for an immune response. A time course of protein expression by western blotting using anti-tPA antibodies will be insightful.

It is generally assumed that mRNA expression is followed by a similar protein expression. It was not the purpose of our study to verify the time course of protein expression, all the more that WB did not work. Overtime protein detection by IP should take into account the natural decay of 35S L-Met that remarkably decreases in a month. It would also be very time consuming, expensive, if not useless.

  1. Presenting an image of a single cell in support of the hypothesis that tPA sequence is essential for expression of L1R and its localization to the membrane is hardly sufficient. Including a field of view with multiple cells and the enlarged single cell image will be more appropriate.

IF images only wanted to prove the evident L1 expression when using the FPtPAL1R recombinant, whereas this expression could not be found when cells were infected with FPL1. By these large images, IF localization can be seen in detail. Conversely, no positive cells were found in Fig. 3 1b, 2b, 3b, where multiple cells are shown.

  1. Figure 1 A & B, label lane named “C” as 5. As labeled, it is confusing. Correct the labeling on line 295.

We did it.

  1. Legend to Figure 3. Line 35: shouldn’t this be (lane 4) and not lane 5?

Yes, it was our mistake. We corrected it.

  1. Line 362: (Fig. 4A; 1a, 1b, 1c). Shouldn’t this be 1a, 2a and 3a?

Yes, we corrected it.

  1. Figure 4: “A” missing from Figure number.

It was inserted again.

  1. Lines 311 and 314: expression “peaked” at 4 days instead of “raised”.

We corrected it. Thanks.

Reviewer 3 Report

Comments and Suggestions for Authors

The manuscript by Radaelli and coworkers sought to characterize  recombinant fowlpox virus expressing the vaccinia virus (VACV) L1R gene encoding sequences with and without the tissue plasminogen activator signal sequence (tPA) as an enhancer for the gene expression. The different recombinant fowlpox virus expressing the VACV-L1R (FP-L1R and FP-L1R_tPA) were characterized for the genetic stability, transgene expression, and protein synthesis. In detail, they characterized the infection of FP-L1R and FP_L1R_tPA in different cells using immunoprecipitation, expression of viral RNA and immunofluorescence.

They confirmed the successful integration of the L1R gene sequences with and without the tPA. In addition, they also isolated mRNA from L1R gene in Vero and MRC-5 cells after different time points. Of note, the mRNA was expressed in Vero and MRC-5 up to 34 days p.i.. In addition, L1 protein synthesis was confirmed by immunoprecipitation in FP-L1R_tPA infected cells. No L1 protein synthesis was detected by immunoprecipitation in FP_L1 infected cells. No L1 protein was detected by Western Blot. They also performed immunofluorescence with and without permeabilization.

In summary, the manuscript is well-written and formatted. However, the amount of data is very limited and the key message is lacking.

The authors should provide more data on the characterization with regard to the differences in the L1R with and without tPA. What is the mechanisms of the tPA for the L1 expression? And why is the L1 without the tPA expressed only intracellularly as seen in the immunofluorescence? And why is there is no L1 protein expression detected by Western Blot analysis? Why is there no L1 synthesis in the FP-L1 infected cells? In addition, the L1-synthesis of the immunoprecipitation is difficult to recognize, so the authors should include new data with clear results also including the appropriate controls.

Another point that needs to be discussed in more detail is the expression of the mRNA for a longer time period in the Vero cells compared to the MRC-5 cells? What is the mechanism? In addition, is it an advantageous characteristic that the mRNA is present for about 34 days in the non-susceptible cells? Does this imply an adverse reaction after using it for vaccination? What is the advantage of presenting the mRNA for longer time to the host immune system? Does this activate an inflammatory response that might result in a cytokine storm? Is all the mRNA transcribed into L1 protein? What is the protective effect of an immune response directed against the L1 protein since there are many other MV-surface proteins that are involved in the entry-fusion complex? Is there a cross-reactivity of the L1-specific immune responses against mpox virus or other orthopoxvirus e.g. cowpox virus or alascapox virus? The authors should include this in the discussion.

In addition, how is the immunogenicity of the FP-L1R and the FP_L1R_tPA vaccine candidate in vivo in mice? Are there any differences for the activation of VACV-neutralizing and binding antibodies?

A minor point is the misuse of the L1R term for the protein since this names the gene while L1 names the protein.

Another minor point is the description of the viruses used in the manuscript. Here the authors describe four FP recombinant viruses: FPL1R, FPA27L, FPA33R, and FPB5R, that expressed the VV L1R, 191 A27L, A33R, and B5R proteins, what is the intention of these viruses expressing other MV and EV-VACV-proteins.

Comments on the Quality of English Language

ok, could be improved

Author Response

The manuscript by Radaelli and coworkers sought to characterize recombinant fowlpox virus expressing the vaccinia virus (VACV) L1R gene encoding sequences with and without the tissue plasminogen activator signal sequence (tPA) as an enhancer for the gene expression. The different recombinant fowlpox virus expressing the VACV-L1R (FP-L1R and FP-L1R_tPA) were characterized for the genetic stability, transgene expression, and protein synthesis. In detail, they characterized the infection of FP-L1R and FP_L1R_tPA in different cells using immunoprecipitation, expression of viral RNA and immunofluorescence.

They confirmed the successful integration of the L1R gene sequences with and without the tPA. In addition, they also isolated mRNA from L1R gene in Vero and MRC-5 cells after different time points. Of note, the mRNA was expressed in Vero and MRC-5 up to 34 days p.i.. In addition, L1 protein synthesis was confirmed by immunoprecipitation in FP-L1R_tPA infected cells. No L1 protein synthesis was detected by immunoprecipitation in FP_L1 infected cells. No L1 protein was detected by Western Blot. They also performed immunofluorescence with and without permeabilization.

In summary, the manuscript is well-written and formatted. However, the amount of data is very limited and the key message is lacking.

  1. The authors should provide more data on the characterization with regard to the differences in the L1R with and without tPA. What is the mechanisms of the tPA for the L1 expression? And why is the L1 without the tPA expressed only intracellularly as seen in the immunofluorescence? And why is there is no L1 protein expression detected by Western Blot analysis? Why is there no L1 synthesis in the FP-L1 infected cells? In addition, the L1-synthesis of the immunoprecipitation is difficult to recognize, so the authors should include new data with clear results also including the appropriate controls.

The low expression level of L1 by the FPL1R recombinant was studied in vivo previously (Bissa et al, Antiviral Res 134, 2016). It did result in a very poor antibody response and suggested the construction of this FPtPA-L1R recombinant. The tissue plasminogen activator signal sequence (tPA) is able to drive the target protein into the cellular secretion pathway as well explained by Kou et al, Immunol Lett 190, 2017 and Ondondo et al. J HIV & AIDS 2.4, 2016. The L1 protein is not carried to the cell membrane and remains in the intracellular compartment where it is probably easily degraded. The enhancement of L1 expression when the recombinant is provided with tPA, as demonstrated by IP and IF, is thus due to the fact that the protein is carried to the cell surface. L1 synthesis also occurs when infecting cells with the unmodified FPL1R recombinant, but L1 expression is very low, almost undetectable, even by the very sensitive IP assay that uses radioactivity. Thus, by remaining in the intracellular compartment, the protein is unable to induce an immune response. By carrying L1 to the membrane surface, tPA may be able to enhance the immunogenicity of a putative vaccine. This is better described now in the Discussion and Conclusions section.

As for the immunoprecipitation assay (IP) is a cleaner, more advanced, and valuable assay than the Western Blot (WB) assay. By WB, proteins loaded on the gel are denatured, whereas by IP the binding of SPA-Ab to the protein occurs before loading the protein unto the gel, as described in the Material and Methods section. Thus, by IP also native proteins are loaded, recognized also for their conformation. IP thus allows the detection of both denatured and non-denatured proteins. However, IP assay requires the use of radioactivity and longer times, it needs to be carefully set up, and therefore it is not used routinely.

Consequently, proteins that may not be recognized by WB, can be recognized by IP. Thus, as the heterologous L1 protein was detected only by IP in the three cell lines infected by the FPtPA-L1R recombinant, this suggests that:

  1. the protein is expressed (which is the aim of our study) not only by CEF where the virus replicates, but also by non-human primate cells (Vero) and human MRC-5 cells, where an abortive replication cycle takes place
  2. if the protein is not detected by the WB assay, this may be due to denatured proteins
  3. in our case, both mouse monoclonal and rabbit polyclonal antibodies were used in WB unsuccessfully. Obviously, we did not show the negative results of our WBs, but shifted to the IP assay, which could give us the chance to detect native proteins, recognized by their conformation
  4. the L1 detection by IP may thus be ascribed to the recognition by the polyclonal rabbit hyper-immune serum of the native conformational form of the protein, which was run in the IP assay, whereas the denatured form of the protein, which was run in the WB assay, could not be detected
  5. the use of commercial antibodies against tPA would not solve the problem. We inserted in our recombinant only the tPA signal sequence (a very short sequence) which is cleaved away in the mature protein. Therefore, anti-tPA antibody could not be used to detect the L1 protein. Moreover, by using an anti-tPA antibody, we would not detect the L1 protein expressed by the unmodified FPL1R recombinant, used for comparisons in lanes 3.
  6. the L1 myristylated envelope protein contains disulfide bonds, formed in the cytoplasm by the virus-encoded disulfide bond formation pathway (Kou et al, Immunol Lett 190, 2017; Ondondo et al. J HIV & AIDS 2.4, 2016), that may be critical for antibody recognition of the protein.
  7. Another point that needs to be discussed in more detail is the expression of the mRNA for a longer time period in the Vero cells compared to the MRC-5 cells? What is the mechanism? In addition, is it an advantageous characteristic that the mRNA is present for about 34 days in the non-susceptible cells? Does this imply an adverse reaction after using it for vaccination? What is the advantage of presenting the mRNA for longer time to the host immune system? Does this activate an inflammatory response that might result in a cytokine storm? Is all the mRNA transcribed into L1 protein? What is the protective effect of an immune response directed against the L1 protein since there are many other MV-surface proteins that are involved in the entry-fusion complex? Is there a cross-reactivity of the L1-specific immune responses against mpox virus or other orthopoxvirus e.g. cowpox virus or alascapox virus? The authors should include this in the discussion.

The longer mRNA expression in Vero cells compared to MRC-5 cells is probably due to the different animal origin and replicative potential of the virus in Vero cells, as we better explained now in the Discussion and Conclusions section. The advantage of this long-lasting expression suggests that proteins will be presented at the cell membrane for a longer time and be recognized by the immune system for long. Thus the immune response may be stimulated longer by this new viral recombinant also in mammalian cells, where a new putative vaccine should be used.

We have planned to use this new construct FPtPAL1R in vivo together with the other three FP recombinants, FPA27L, FPA33R, and FPB5R, that express the A27, A33, and B5 proteins, respectively, to induce immunogenicity both against EV and MV viral particles. In particular, L1 protein plays a significant role in infection and morphogenesis, as it is involved in cellular entry, and it is the target of neutralizing antibodies. In the absence of tPA, we previously demonstrated that L1 protein was almost unable to elicit an antibody response. We thus decided to construct this new recombinant by adding tPA with the aim at increasing the immune response. L1 is also well-conserved in all orthopoxviruses and it shows 98,8 % similarity with the M1 MPXV ortholog (Gao et al, Virology J. 20, 2023). We included this information in the Introduction and Discussion and Conclusions section.

  1. In addition, how is the immunogenicity of the FP-L1R and the FP_L1R_tPA vaccine candidate in vivo in mice? Are there any differences for the activation of VACV-neutralizing and binding antibodies?

Preliminary results show a clear enhancement of the antibody response and neutralizing activity by this new FPtPA-L1R recombinant compared to the response obtained when using the unmodified FPL1R.

  1. A minor point is the misuse of the L1R term for the protein since this names the gene while L1 names the protein.

The names of the genes and proteins has now been corrected all over the manuscript.

  1. Another minor point is the description of the viruses used in the manuscript. Here the authors describe four FP recombinant viruses: FPL1R, FPA27L, FPA33R, and FPB5R, that expressed the VV L1R, A27L, A33R, and B5R proteins, what is the intention of these viruses expressing other MV and EV-VACV-proteins.

The mentioned 4 FPL1R, FPA27L, FPA33R, and FPB5R recombinants were used in a previous in-vivo study and in a recent still-unpublished study where these immunogens were employed in different prime-boost regimens. These recombinants are the most representative ones for protein expression of EV (A33 and B5) and MV proteins (L1 and A27), all well conserved among OPVXs. As protein expression by the unmodified FPL1R was very low, we thought of enhancing its expression by the FPtPA-L1R recombinant where L1 could be driven to the cell surface and expressed it at higher level thus improving its immunogenicity and the efficacy of a putative vaccine.

  1. Comments on the Quality of English Language ok, could be improved

Round 2

Reviewer 2 Report

Comments and Suggestions for Authors

Although the authors provide arguments why the wetern blotting using anti-L1R antibodies did not work, choosing not to use the tPA antibodies appears short sighted, especially given the authors see two bands of size 21 and 29 kDa, which could likely be the processed and unprocessed protein respectively. If this is the case, the authors would see only the 29 kDa unprocessed band if probed using anti-tPA antibody. Did the authors consider this possibility? 

The entire point of this manuscript is to establish an expression system to make vaccine and thus identifying conclusively the product of this expression system is critical for this process.

Author Response

Comments and Suggestions for Authors

Although the authors provide arguments why the western blotting using anti-L1R antibodies did not work, choosing not to use the tPA antibodies appears short sighted, especially given the authors see two bands of size 21 and 29 kDa, which could likely be the processed and unprocessed protein respectively. If this is the case, the authors would see only the 29 kDa unprocessed band if probed using anti-tPA antibody. Did the authors consider this possibility? 

May be we were not clear enough in answering to this question. We did not add the tPA ahead of L1R gene, but only the signal sequence of tPA (that in our manuscript was called simply tPA, as first indicated in the Introduction section), which is going to be lost in the protein. Thus, although we considered this possibility, there are many reasons for not using an anti-tPA antibody:

  1. The signal sequence is very short (63 nt, 21 AA), whereas the tPA protein consists of 562 AA. This is also the reason why in Fig. 1, when we wanted to be sure that tPA was present in the new FPtPA-L1R construct we did not amplify only the gene (which is too short) but we amplified a part of the signal sequence of tPA and a part of the L1R gene;
  2. In our IP assay we also wanted to compare the L1 protein expressed by the unmodified FPL1R construct (lanes 3) with the L1 protein expressed by the modified FPtPAL1R construct (lanes 4). If we used anti-tPA antibodies, we would never recognize the protein of the unmodified FPL1R construct that does not possess it, and could not make any comparison;
  3. the signal sequence gets lost when the protein is expressed.

New corrections were added and highlighted in blue to clear this point.

The entire point of this manuscript is to establish an expression system to make vaccine and thus identifying conclusively the product of this expression system is critical for this process.

Our main goal was to verify the expression of the protein by the modified FPtPAL1R construct and whether the expression was increased by the insertion of tPA. Both IF and IP demonstrate that the L1 protein expression was definitely increased due to the presence of tPA signal sequence, which is responsible for the translocation of the protein to the membrane surface. That would favor its presentation to and recognition by the immune system. The construct may thus be more effective for vaccination.

Reviewer 3 Report

Comments and Suggestions for Authors

To further show an advantageous effect of the tPA-L1R and to confirm the manuscript's main conclusion, preliminary data on antibody responses induced by this new FPtPA-L1R recombinant compared to the response obtained when using the unmodified FPL1R. should be included within the manuscript.

Author Response

Comments and Suggestions for Authors

To further show an advantageous effect of the tPA-L1R and to confirm the manuscript's main conclusion, preliminary data on antibody responses induced by this new FPtPA-L1R recombinant compared to the response obtained when using the unmodified FPL1R should be included within the manuscript.

The main goal of this study was to verify the expression of the protein by the new FPtPA-L1R construct and whether the L1 transgene expression could be increased by the insertion of the tPA signal sequence. Both IF and IP demonstrated that the L1 protein expression was definitely increased by the use of the modified FPtPAL1R construct compared with the unmodified FPL1R construct. That would favor its recognition by the immune system. The construct may thus be used for vaccination and might be more successful than the unmodified construct. The immune response for the unmodified construct was previously verified in vivo and was very low (as mentioned in the manuscript, Bissa et al. 2016, Antiviral Res) and prompted us to the construction of a more efficient recombinant, able to express the protein on the cell membrane. The in-vivo study for the detection of the immune response, including the humoral response, is foreseen by a further more articulate study, where different immunization protocols will be evaluated in adequate animal models.

Round 3

Reviewer 2 Report

Comments and Suggestions for Authors

I think the manuscript is sufficiently revised to be acceptable for publication.

Reviewer 3 Report

Comments and Suggestions for Authors

I think the manuscript is sufficiently revised to be acceptable for publication.